# Deep learning identifies histopathologic changes in bladder cancers associated with smoke exposure status

**Okyaz Eminaga**[1]*, **Hubert Lau**[2,3], **Eugene Shkolyar**[4], **Eva Wardelmann**[5], **Mahmoud Abbas**[5]*

1 AI Vobis, Palo Alto, California, United States of America, 2 Department of Pathology and Laboratory Medicine, Veterans Affairs Palo Alto Health Care System, Palo Alto, California, United States of America, 3 Department of Pathology, Stanford University School of Medicine, Palo Alto, California, United States of America, 4 Department of Urology, Stanford University School of Medicine, Palo Alto, California, United States of America, 5 Department of Pathology, University Hospital of Muenster, Münster, Germany

* okyaz.eminaga@aivobis.com (OE); Mahmoud.Abbas@ukmuenster.de (MA)

**Data Availability Statement:** We provided the data (external_validation_set.csv) used for downstream analyses and we refers the readers to the PLCO study organizer to obtain the raw data after

## Abstract

Smoke exposure is associated with bladder cancer (BC). However, little is known about whether the histologic changes of BC can predict the status of smoke exposure. Given this knowledge gap, the current study investigated the potential association between histology images and smoke exposure status. A total of 483 whole-slide histology images of 285 unique cases of BC were available from multiple centers for BC diagnosis. A deep learning model was developed to predict the smoke exposure status and externally validated on BC cases. The development set consisted of 66 cases from two centers. The external validation consisted of 94 cases from remaining centers for patients who either never smoked cigarettes or were active smokers at the time of diagnosis. The threshold for binary categorization was fixed to the median confidence score (65) of the development set. On external validation, AUC was used to assess the randomness of predicted smoke status; we utilized latent feature presentation to determine common histologic patterns for smoke exposure status and mixed effect logistic regression models determined the parameter independence from BC grade, gender, time to diagnosis, and age at diagnosis. We used 2,000-times bootstrap resampling to estimate the 95% Confidence Interval (CI) on the external validation set. The results showed an AUC of 0.67 (95% CI: 0.58–0.76), indicating non-randomness of model classification, with a specificity of 51.2% and sensitivity of 82.2%. Multivariate analyses revealed that our model provided an independent predictor for smoke exposure status derived from histology images, with an odds ratio of 1.710 (95% CI: 1.148–2.54). Common histologic patterns of BC were found in active or never smokers. In conclusion, deep learning reveals histopathologic features of BC that are predictive of smoke exposure and, therefore, may provide valuable information regarding smoke exposure status.

closing a data sharing agreement with the study organizer (https://prevention.cancer.gov/major-programs/prostate-lung-colorectal-and-ovarian-cancer-screening-trial-plco).

**Competing interests:** The authors have declared that no competing interests exist.

## Introduction

Bladder cancer (BC) is the sixth most common malignancy in the United States, with an estimated 83,730 new diagnoses in 2021 [1]. Smoking is a well-accepted risk factor for developing BC, accounting for up to 50% of cases [2]. The association between BC and tobacco consumption [1, 3] is attributable to aromatic amines and heterocyclic compounds [4] that incite the carcinogenic molecular mechanisms in bladder urothelium [5]. Previous studies reported a potential correlation between histopathologic features and smoke exposure in BC [6, 7] as a consequence of its carcinogenic mechanisms. Additionally, intensity of cigarette consumption and timing of cessation have been linked to differing histopathologic subtypes and disease aggressiveness [8–11]. However, the existing data from previous studies are characterized by inconsistencies and a focus on the controversial association between BC grade and smoke exposure. As a result, it remains unclear whether there is an association between specific histologic features in BC and smoke exposure. Given this knowledge gap, the current study investigated the potential association between histology images and smoke exposure status.

To further characterize the relationship between histopathology and smoke exposure, the current study used deep learning (DL) to examine histologic images of BC and investigated the predictive value of histopathology for smoke exposure status. DL is a broad family of machine learning methods in artificial intelligence (AI). DL incorporates deep convolutional neural networks (CNN) [12, 13], which have demonstrated exceptional utility in computer vision due to strong performance in pattern recognition tasks such as histopathology[14–16] or treatment response prediction from histology images [17, 18]. Herein, we investigate the histopathological features associated with smoke exposure and characterize this relationship using DL. We developed a CNN model that predicts, scores and extracts latent features associated with smoke exposure using histologic images of BC from the multi-center Prostate, Lung, Colorectal, and Ovarian (PLCO) Cancer Screening Trial.

## Material and methods

### Study cohort

The PLCO Cancer Screening Trial is a multi-center, randomized trial designed to evaluate the impact of screening on cancer-related outcomes [19, 20]. Although this trial did not screen for BC, it tracked diagnoses of BC during the trial period. Briefly, 154,900 participants from the general population aged 55 through 74 were enrolled between 1993 and 2001 [19]. Only subjects without a history of prostate, lung, colorectal, or ovarian cancer were enrolled. After enrollment, participants were randomized to "screening" or "no-screening" arms for these malignancies and followed through the end of 2008. Smoke exposure was regularly documented without restricting smoking exposure levels as part of an annual survey via mailed questionnaires during the follow-up period [19]. Cancer diagnoses were confirmed by retrieving results and information from medical records and the cancer registry system. During the study follow-up period, 1,430 cases of BC were diagnosed.

For model development, in-training validation, and external validation, 285 cases were available from 9 centers (University of Colorado, Georgetown University, Pacific Health Research and Education Institute -Honolulu-, Henry Ford Health System, University of Minnesota, Washington University in St Louis, University of Pittsburgh, University of Utah, and Marshfield Clinic Research Foundation). A total of 483 paraffin-embedded hematoxylin and eosin (H&E)-stained whole slide images (WSI) captured at 20x magnification using a Leica Biosystem Scanner (Wetzlar, Germany) were included. All samples were originally obtained through transurethral resection of bladder tumors. S1 Fig provides the flowchart for the study cohort.

## Data preprocessing

The development dataset consisted of 90 WSI of 55 cases from a single center (Marshfield Clinic Research Foundation) for training and 26 WSI of 16 cases from another center (Georgetown University) for model optimization (in-training validation). A total of 367 WSI from 214 cases from the remaining study centers were available for the external validation set. From there, we limited the validation dataset to cases selected either from never-smokers or active smokers, for a total of 155 WSI from 94 cases diagnosed with BC. The purpose of excluding former smokers from external validation was to eliminate the potential effect of smoking cessation on histopathological appearance and, consequently, on downstream analyses.

Given the distinctive hematoxylin-stained nuclei and contrasting appearance of urothelium from less cellular stromal tissue, we applied a color mask to the WSI thumbnails, whose pixels have hue, saturation, and value (HSV) colors located in the color range between (H:140, S:20, V:50) and (H:150, S:255, V:255) (**S2 Fig**). Background noise was then filtered from the WSI using the erosion function (kernel size: 2 x 2). Masked areas were expanded, and missing portions were filled with the dilation function (kernel size: 5 x 5) provided by the OpenCV library [21]. Masked regions were split into 10% overlapping squares, rescaled, and remapped to the original WSI size. The masked areas were tiled on WSI at 10x magnification into small patches (512x512 pixels, one pixel = 1 μm) for processing due to memory constraints. The red, green, and blue (RGB) color space was the default color for model training and evaluation. All patches were labeled by one of three cigarette exposure statuses: "never smoker," "active smoker," and "former smoker" during the follow-up period up to BC diagnosis.

## Model development

Fig 1 summarizes the study's workflow for model development. All models were trained to classify patches into three classes of smoke exposure (i.e., never smoker, former smoker, and active smoker) to facilitate learning distinguishable patterns between different smoke exposure conditions.

Neural architecture search (NAS) with a predefined search space was employed with the training set [22] to determine optimal model morphology. PlexusNet was used to perform a grid search to choose the path, width, and depth of the model architecture [23] (S3 Fig). The grid search was preferred because the search space for architecture hyperparameters was finite, limited, and searchable using the conventional grid search. The computation cost for grid search was also reasonable. The following parameters define the search space.

$$w \in \{2, 3, 4\};$$

$$d \in \{3, 4, 5, 6\};$$

$$c \in \{4, 6, 8\};$$

$$b \in \{\text{soft attention}, \text{inception}, vgg, resnet\};$$

$$j \in \{2, 3\}$$

where $w$ is the graph branching factor, $d$ is the graph level depth, c is the initial channel number used to populate the channels in the first and consequence blocks, b is the block type, and j is the junction number between two paths (root and auxiliary paths). The hyperparameters for

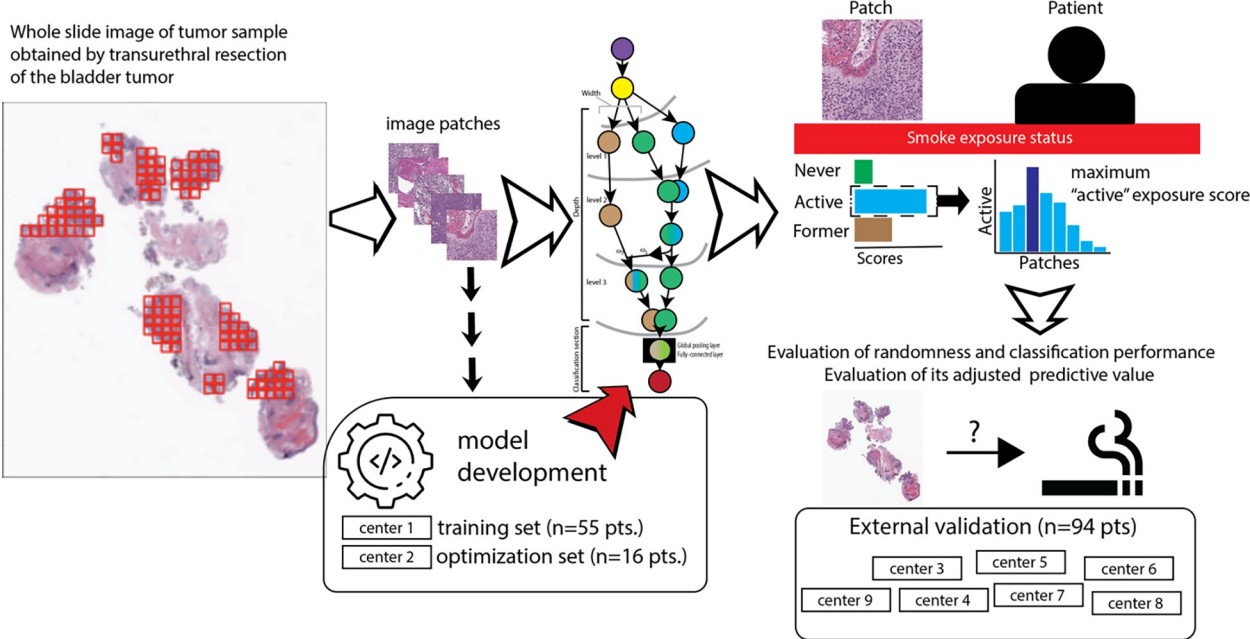

**Fig 1. Summarizes the workflow of model development and external validation as well as the processing of the whole slide image to obtain the exposure score for each patient.** The whole slide images are tiled in small patches; the image patches from training or optimization, aka in-training validation set, were used to develop and optimize the model for predicting the smoke exposure status (never, active, and former smoker). After model development, the pipeline processed the whole slides and estimated the smoking exposure status for each patient. Here, the exposure scores derived from the confidence scores of the "active smoker" category are first calculated for all image patches per patient and then assigned the maximum exposure score to the corresponding patient.

model training were prefixed and included the ADAM optimization algorithm [24] with a learning rate of 1e-4 (Beta1: 0.9 and beta2: 0.999 by default), the categorical cross-entropy loss function, and a batch size of 32 patches. The evaluation metrics were the area under the curve (AUC) of the receiver operating characteristic (ROC) curve and the classification accuracy (CA). Image augmentation was applied with 50% probability and generated a variety of patches to increase the likelihood of a generalizable training set and to regulate the model fitness. Image augmentation included random rotation (angle range: [–40,40]), manipulation of brightness (0.1), image resolution quality (i.e., JPEG image compression rates between 80 and 100%), color saturation (manipulation of HSV color space), and image contrast, as well as the random horizontal flip. Using these search parameters and parallel computing, NAS generated and examined 288 models in 48 hours on a single GPU card; the optimal model was determined according to the highest classification accuracy by 2-fold cross-validation after 10 training epochs.

After identifying the optimal model architecture, it was trained using the training set at the patch level, using the same augmentation and optimization configuration until model convergence. Model performance was evaluated per epoch using the in-training validation data set. The summation of AUC and CA on the in-training validation set was used to determine the best model at the patch level.

We estimated and evaluated the exposure scores on the external validation set to determine the histopathologic association between "exposure score" and active smoker status. To identify cases with active cigarette smoke exposure from histology images, the maximum confidence score of the "active smoker" class generated from the available patches was applied to the corresponding case. This maximum confidence score is herein labeled as the "exposure score" of

the corresponding case. Afterward, the median exposure score was estimated from the development data set and used as the threshold for conversion of the exposure score from a continuous to binarily categorized parameter, and this was locked as the threshold for the external validation set.

To investigate the effect of the magnification, we repeated the previous steps (except the NAS step; here, we used the resulting model architecture from the first evaluation) to develop and validate a prediction model on images at 20x magnification. After fixing the optimal magnification level, we also examined the attention-based multi-instance learning defined according to Ilse et al. [25], as it has shown performance benefits for certain tasks in digital pathology [26, 27]. The backbone model was the PlexusNet model we described earlier.

## External validation

The external validation data set was used to determine the discriminatory accuracy and randomness of the exposure score for smoking status. Randomness was estimated using AUC [28]. Discriminatory accuracy was determined by classification accuracy, specificity, sensitivity, f1-score, precision, negative and positive predictive values. The contribution of demographic and pathologic variables such as age, gender, BC grade, and time to diagnosis were also evaluated. The variable 'time to diagnosis' represents the duration of the participant's inclusion in the screening study. Two mixed-effect regression models were built to assess the independent variables to predict smoke exposure. The base model considered gender and age at diagnosis as random effects and BC grade and time to diagnosis as fixed effects. Exposure scores were included as a continuous parameter for the first model and a binary categorization for the second model. The general equation of the mixed effect regression model is:

$$\hat{y}_{(smoke\ status)} = Exposure\ score \bullet BC\ grade + Time\ to\ diagnosis + (gender||age\ at\ diagnosis)$$

The symbols represent interaction (.), conditional relations (||), and grouping (()) between different variables.

The random effect incorporated gender and age to consider the inherent variability across the population. We designated time for diagnosis and malignancy grade as fixed effects. This decision aligns with the prospective nature of the PLCO study, where the time to diagnosis was meticulously recorded. We postulate that the time to diagnosis encompasses the latent period leading up to the clinical manifestation of bladder cancer. Furthermore, we suggest that the malignancy grade correlates with the smoking status of the individuals in the study. We defined the interaction between exposure score and BC grade to avoid a singular model fit resulting from the limited random effect. Moreover, we assumed that BC grade and our exposure score are interacted given their information source (i.e., histology appearance).

In parallel, we applied t-SNE (t-distributed Stochastic Neighbor Embedding) to visualize the 3D space of the latent features resulting from the global pooling directly after the last convolutional layer of the model for all patch images [29]. Here, we labeled the data points according to the decile of the exposure score and the smoking status. Then, we searched for subspaces either dominated by active smokers or never smokers where their decile exposure score gradient is correlated with the smoking status. Here, the boundary definition for each subspace was explored on the 3D data points and fixed when at least 80% of a minimum of 200 patch images originated either from never-smoker or active-smoker patients. Finally, the patch images were clustered to identify similar patterns for each subspace, and we calculated the overall case number of active or never smokers in each space.

Moreover, two genitourinary pathologists (MA and HDL) independently examined the image clusters for each subspace. They provided their overall impression of diagnosis and

heterogeneity for each subspace and reported other notable findings. Both pathologists were blinded to the clinical information, and no time limitation for the examination was specified.

### Statistical analyses

All analyses were performed using 2,000-times bootstrap resampling to estimate the 95% confidence interval (CI) on the external validation set. A t-test was used for mean comparison between the two groups, and the non-parametric Kruskal–Wallis test was used for distribution evaluation between multiple groups. The reported $p$-values are two-tailed, and statistical significance was considered when $p < 0.05$. The false discovery rate was applied for test comparison, and the statistical significance was considered when FDR $< 0.1$. The statistical robustness of AUC was estimated using the statistical Power. We set the decision threshold for a robust statistical Power to 80% at an alpha level 0.05 [30–34].

### Software and hardware configuration

Statistical analysis was performed using Python 3.8 (Python Software Foundation, Wilmington, DE) and R 4.0.3 (R Foundation for Statistical Computing, Vienna, Austria). The Keras library [35], a high-level wrapper of the TensorFlow framework (2.5), and Scikit-learn [36] were used to develop the models, and "lmer4" was used for mixed-effect regression modeling [37]. Scikit-image (0.17.2), OpenCV (4.10), and matplotlib (3.3.2) were used for image processing and plotting. All analyses were performed on a GPU machine with a 44-core Intel processor with 64 GB RAM (Intel, Santa Clara, CA), from which an 8 GB virtual storage memory was allocated for fast read/write access to temporarily store patch images, 10 TB hard disks to store the whole slide images, and a single NVIDIA 1080 ti GPU with 11 GB VRAM.

### Results

A total of 10,406 patches were generated for the training set, 3,272 patches for the in-training validation set, and 17,416 for the external validation set. **Table 1** summarizes the cohort description.

The final model architecture was determined (architecture configuration: d = 5; w = 2; j = 3, c = 6; b = soft attention) to generate exposure scores for the external validation set. The parameter capacity of the final model was 272,743 parameters, and the first fully connected layer, or the global pooling layer, had 64 dense units. The exposure scores of BC cases were equally distributed between the seven centers (Kruskal-Wallis chi-squared = 4.471, degree of freedom = 6, $p = 0.6132$), indicating no impact of the center of diagnosis or variation in H&E staining protocols on the exposure scores.

The exposure score achieved an AUC of 0.64 (95% CI: 0.53–0.76) for distinguishing never-smokers from active smokers in the external validation set (**Fig 2**). As a binary category using a threshold of 65, the exposure score provided a balanced accuracy of 68% with an AUC of 0.67 (95% CI: 0.58–0.75), a sensitivity of 82% (95% CI: 71–93) and specificity of 51% (95% CI: 37–65). The F1-score was 0.70 (95% CI: 0.59–0.79) with a precision of 0.61 (95% CI: 0.49–0.72). Table 2 shows the confusion matrix for smoke exposure status. The statistical power of the study was determined to be 82.2%, surpassing the commonly accepted threshold of 80% at an alpha level of 0.05.

When comparing models trained on histology images at 10x magnification versus those trained on images at 20x magnification, we observed performance degradation on the external validation set (AUROC: 0.528; 95% CI: 0.432–0.624), indicating that magnification does impact performance. In addition, MIL failed to provide a non-random prediction for the smoke exposure status (AUROC: 0.561; 95% CI: 0.437–0.685).

**Table 1. Describes the cohort characteristics of the current study.**

|  | All | Training set | In-training validation set | External validation set |
|---|---|---|---|---|
| **Patients, n (%)** | 165 (100) | 55 (42.86) | 16 (12.38) | 94 (44.76) |
| **Age at diagnosis, years, median (IQR)** | 65 (60–69) | 63 (58–68) | 69 (65–71) | 63 (60–67) |
| *Gender, n (%)* |  |  |  |  |
| Male | 125 (75.8) | 44 (80.0) | 13 (81.3) | 68 (72.34) |
| Female | 40 (24.2) | 11 (20.0) | 3 (18.7) | 26 (27.66) |
| *Smoking status* |  |  |  |  |
| Never smoker | 63 (38.2) | 9 (16.4) | 5 (31.2) | 49 (52.13) |
| Active smoker | 57 (34.5) | 11 (20.0) | 1 (6.3) | 45 (47.87) |
| Former smoker | 45 (27.3) | 35 (63.6) | 10 (62.5) | 0 |
| *Staging, n (%)* |  |  |  |  |
| Stage 0 | 103 (62.4) | 37 (67.3) | 13 (81.3) | 53 (56.4) |
| Stage I | 33 (20.0) | 11 (20.0) | 2 (12.5) | 20 (21.3) |
| Stage II | 18 (10.9) | 4 (7.3) | 1 (6.2) | 13 (13.8) |
| Stage III | 4 (2.4) | 2 (3.6) | 0 | 2 (2.1) |
| Stage IV | 1 (0.6) | 0 | 0 | 1 (1.1) |
| Unknown | 6 (3.6) | 1 (1.8) | 0 | 5 (5.3) |
| *BC grade, n (%)* |  |  |  |  |
| I (Well Differentiated) | 49 (29.70) | 16 (29.1) | 9 (56.3) | 24 (25.5) |
| II (Moderately Differentiated) | 38 (23.03) | 19 (34.55) | 4 (25.9) | 15 (16.0) |
| III (Poorly Differentiated) | 70 (42.42) | 19 (34.55) | 3 (18.8) | 48 (51.0) |
| IV (Undifferentiated) | 2 (1.21) | 1 (1.82) | 0 | 1 (1.1) |
| Unknown | 6 (3.64) | 0 | 0 | 6 (6.4) |
| **Time to diagnosis, years, median (IQR)** | 8 (4.67–11.67) | 7.33 (4.25–11.12) | 8.75 (5.67–11.83) | 8.33 (5.25–10.75) |
| **Cigarette exposure duration for active smokers in years, mean (95% CI)** | 44 (42–46) | 46 (42–49) | n.c. | 44 (41–46) |
| Pack-year, mean (95% CI) | 55 (47–62) | 60 (34–86) | n.c. | 54 (46–61) |
| *Number of patches* |  |  |  |  |
| Never smoker | 12,641 (39.37) | 2,008 (19.30) | 819 (25.02) | 9,814 (53.24) |
| Active smoker | 11,106 (34.59) | 2,485 (23.88) | 3 (0.09) | 8,618 (46.76) |
| Former smoker | 8,363 (26.04) | 5,913 (56.82) | 2,450 (74.85) | 0 |

CI: Confidence interval; IQR: interquartile range; n.c.: not calculable. BC: bladder cancer. BC grade was judged by the criteria definition of the PLCO study.

The average smoking intensity measured by pack-year (PY) and duration (years) was higher for active smokers with exposure scores above 65 compared to active smokers with exposure scores equal to or below 65 (54.8 vs. 47.7 for PY; 44.4 vs. 40.3 for smoking duration in years). However, these were not statistically significant ($p = 0.5783$ for PY; $p = 0.324$ for smoking duration).

The multivariate mixed-effect logistic regression model identified exposure score as the only predictor for active smoking after adjusting for gender, age at diagnosis, time to diagnosis, and BC grade (Table 3).

We visualized the 3D space of the latent features for the whole patch images in the external validation set using t-SNE (Fig 3). In the spherical 3D space with an uneven surface, we identified one subspace dominated by never-smokers and two feature subspaces dominated by active smokers. The first latent feature space had 820 patch images from 48 patients; of those, 697 patch images (85%) were from 41 never-smokers (85%). The second subspace was defined by the latent features of 217 patch images originating from 52 patients; of those, 173 patch images (80%) were from all 45 active smokers (87%). The third latent feature subspace included 214

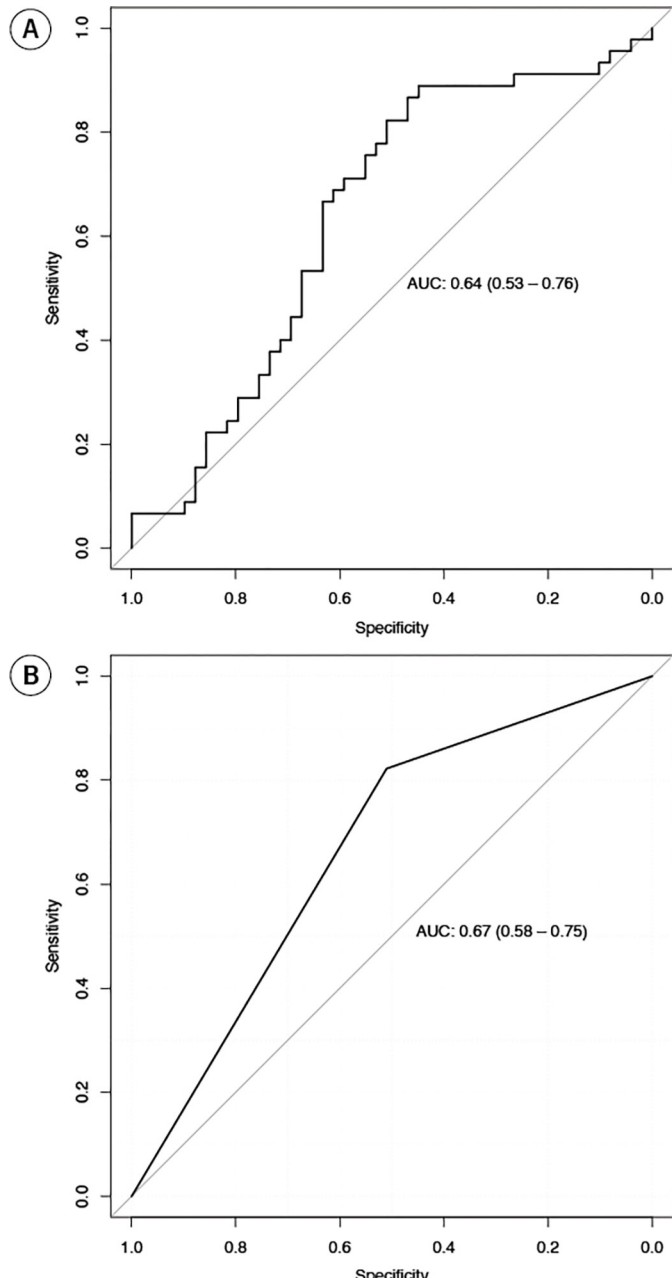

**Fig 2.** The area under the Receiver Operating Characteristic curve (AUC) for the exposure score as (A) a continuous parameter and (B) as a categorized parameter at a threshold of 65.

**Table 2. Confusion matrix for the smoke exposure status determination.**

| Smoke exposure status | | Predicted | |
|---|---|---|---|
| | | Low exposure score, n (%) | High exposure score, n (%) |
| Ground truth | Never smoker, n (%) | 25 (51.0) | 24 (49.0) |
| | Active smoker, n (%) | 8 (17.8) | 37 (82.2) |

**Table 3. Multivariate mixed-effect regression analyses.**

| Parameters | Odd ratio (95% CI) | FDR |
|---|---|---|
| Exposure score (continuous) | 1.009 (1.002–1.02) | 0.044 |
| BC grade | 1.088 (0.942–1.26) | 0.259 |
| Time to diagnosis | 0.998 (0.996–1.00) | 0.208 |
| Exposure score • BC grade | 0.844 (0.668 – 1.07) | 0.208 |
| Exposure score (categorized)++ | 1.710 (1.148–2.54) | 0.044 |

++ originated from a base model with a categorized exposure score. FDR: false discovery rate.

patch images from 22 patients, from which 204 patch images (95.3%) originated from 19 smokers (86%). **Fig 3** provides representative patch images for these subspaces, whereas the complete patch images for all subspaces are online through the external links provided in S1 File.

Two genitourinary pathologists evaluated the subspaces, which revealed trends in BC grade and heterogeneity, as shown in **Fig 4** and **S1 Table**.

## Discussion

Chronic tobacco exposure leads to chemical irritation and chronic inflammation in many organs, including the urinary bladder [38–40]. Moreover, tobacco exposure is a well-known risk factor for BC [41]. We utilized AI to examine whether there is an association between smoke exposure and histopathologic changes in BC. We found that histopathologic features are indeed independent predictors for smoke exposure in BC and that histopathologic changes resulting in distinctly clustering latent features were specific for smoke exposure status, comparable to work done by Auerbach et al. in 1989, which showed histopathologic changes in 88% of BC with similar smoke intensity as our cohort [6].

Although the widespread presence of common histologic features among cases is expected, the 3D feature exploration recognizes three distinctive feature subspaces according to smoke exposure status. Further evaluation of these subspaces unveils trends in malignancy degree (i.e., BC grade and stage) and tumor heterogeneity across these feature subspaces. This observation implies that predictable alterations in the tissue morphology of BC conceivably exist based on smoke exposure status, especially when comparing active smokers to never-smokers. The histologic changes may be associated with the carcinogenic effects of cigarette smoke, which impairs DNA-repair activity [42, 43] and causes epigenetic changes [8], epithelial-mesenchymal transition [44, 45], and hypoxic changes [46], consequently altering urothelial cell [47] and tissue morphology [48, 49].

A feature space dominated by active smokers included a subset of images with keratinizing squamous metaplasia, an uncommon, potentially pre-malignant condition resulting from chronic irritation of the urothelium [50, 51]. Although multiple studies have associated the development of squamous metaplasia of the respiratory tract with active smokers [52, 53], such an association is not well established for squamous metaplasia of the urinary bladder [54]. Studies of risk factors for bladder cancer found cigarette exposure increases the risk for squamous cell carcinoma of the bladder, a rare subtype of bladder cancer that may arise from underlying squamous metaplasia [55]. These findings suggest the need for further investigation of the association between squamous metaplasia of the bladder and smoke exposure.

Epidemiologic studies have shown contradicting results regarding the association between smoking status and BC grade. Some studies find an increased risk for high-grade non-muscle invasive bladder cancers in active smokers [8, 9, 52], whereas other studies did not find this

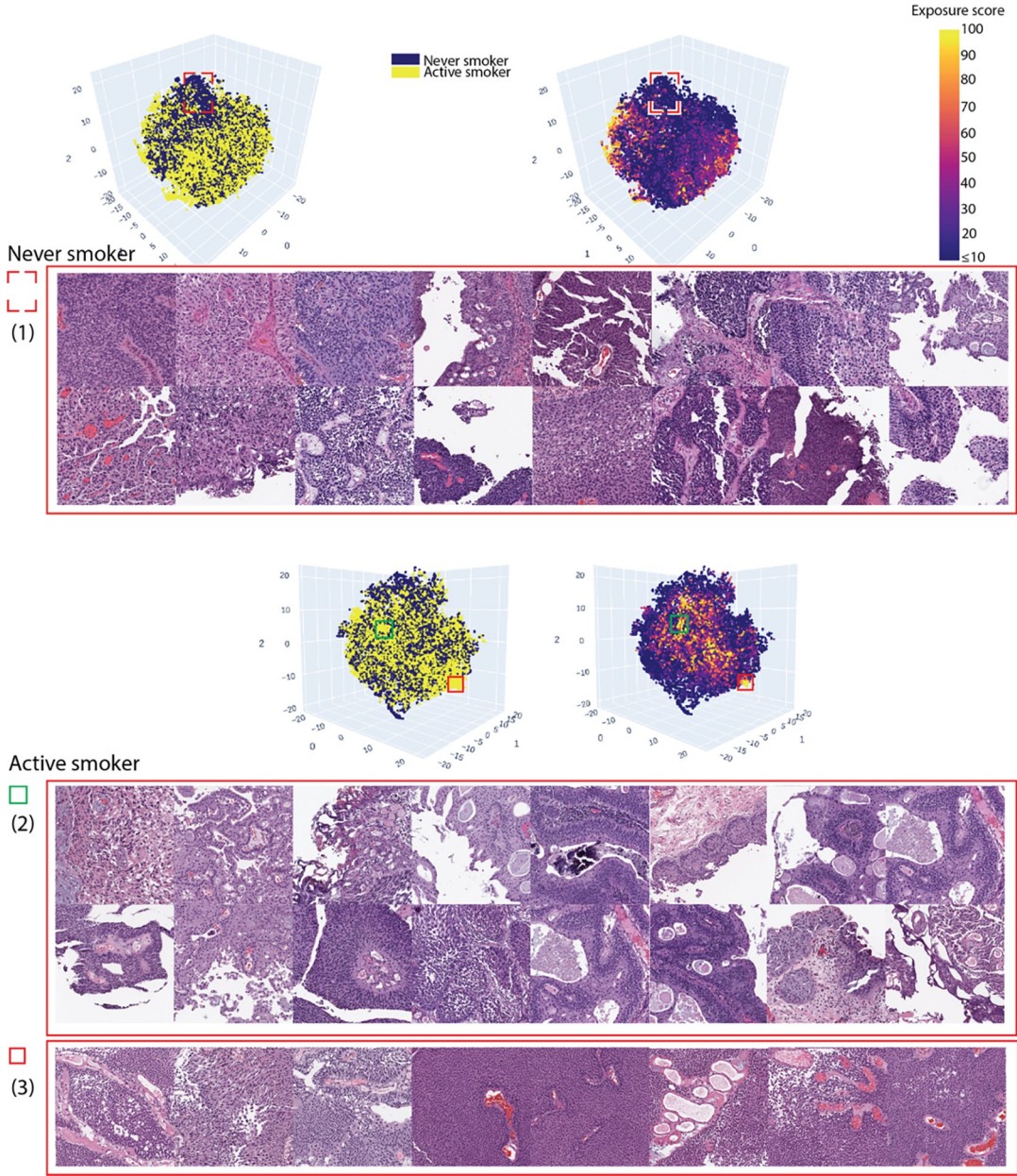

**Fig 3. Reveals the 3D latent feature representation of the external validation set using t-SNE.** We identified three subspaces that met the criteria described in the Material and Method section and highlighted them by bounding boxes. Representative patch images are provided for each subspace. 1) the first latent feature space has a total of 820 patch images from 48 patients; of those, 697 patch images (85%) are from 41 never-smokers (85%); 2) the second subspace is defined by the latent features of 217 patch images that are originated from 52 patients; of those, 173 patch images (80%) are from all 45 active smokers (45 of 52: 87%); 3) the third latent feature subspace includes a total of 214 patch images from 22 patients; of those, 204 patch images (95.3%) are from 19 active smokers (86%)—the patch scale: ~512μm at 100× objective magnification. S1 File provides the clustered patch images for each subspace after stratifying by smoking status (active vs. never smoker). S1 Video provides a 3D navigation through the feature space labeled by the tobacco exposure status.

association with active smokers [8, 56]. Another study led by Sturgeon et al. reported that active smokers with BC diagnosis are rather associated with low-grade BC [57]. The current study did not find any significant association between smoking status and BC grade. One possible explanation is the cancer dynamics of BC in active smokers results in both low- and high-

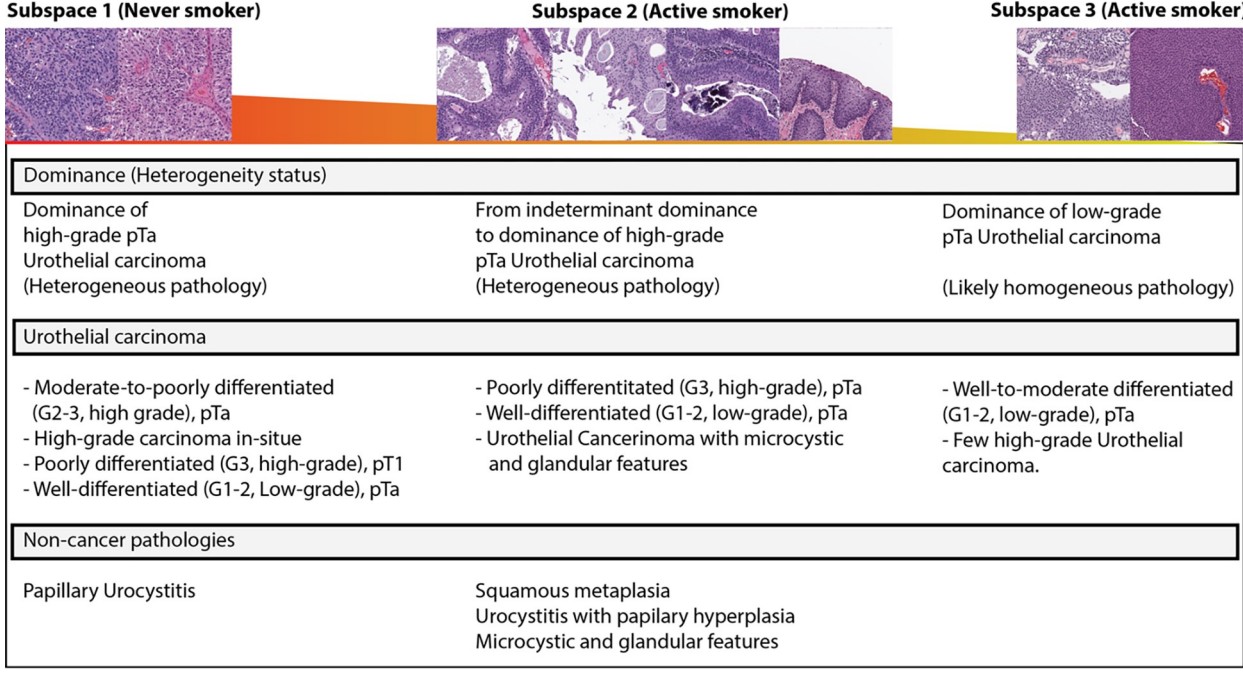

**Fig 4. Summarizes the overall impressions of the subspaces made by two genitourinary pathologists.** Each subspace is dominated either by never-smokers or active smokers.

grade tumors. Interestingly, the feature subspaces dominated by active smokers in this study showed different dominant tumor grades, mirroring the contradictory results in the literature.

Our analyses indicate that small models developed on a relatively limited data set are sufficient to transform histopathologic changes into exposure scores and latent features to investigate the association between these changes and smoking exposure. Other benefits of having a small model compared to existing large models include a lower risk for overfitting on limited datasets due to the low parameter capacity [58] and a smaller representation feature dimension (64 globally pooled features for our model vs. 1,280 globally pooled features for EfficientNet B0 [59]), lower memory occupation (3MB for our model vs. 38 MB for EfficientNet B0 [59]), and a faster image processing (i.e., at least the half of the time of large state-of-art models). Furthermore, our analysis of exposure scores for cases originating from different centers did not show any biases despite each center's having its own slide processing and staining protocol. Moreover, we identified common patterns in multi-site cases using the latent features generated by the model and a well-accepted data visualization algorithm. Accordingly, applying additional stain color normalization is unnecessary, and our image augmentation strategy is sufficient to achieve a generalizable result.

We disregarded the classification prediction for former smokers and only considered the prediction score for active smokers for the association analyses. The primary reason for including the class for former smokers during model development is to regularize the model prediction. However, considering the former smoker's status for external validation is not appropriate given the difficulty in identifying the regeneration level of the tissues resulting from stopping smoke exposure. In contrast, we can assume that the baseline environmental exposure (including passive smoking) affects both smokers and non-smokers equally, while smokers have direct exposure to tobacco.

The association between histology images and smoke exposure status at 20x magnification is random, whereas the association at 10x magnification is significant and not random. This

finding indicates that histologic patterns (i.e., tissue architecture) at 10x magnification are predictive of smoke exposure, supporting the key message of the current work regarding the potential association between active smoke exposure and histological images.

In the current study, we investigated the local features using 3D t-SNE and visualized the 3D feature spaces. We identified global feature subspaces that revealed a high concentration of patches dominated by either active smokers or non-smokers. These findings further indicate the reorganization of histologic features associated with these subspaces. Our pathologists investigated and confirmed distinguishable histologic patterns between the subspaces we identified using 3D t-SNE visualization. We emphasize that the 3D visualization of feature spaces provides significantly more information than the 2D visualization.

Although the current study provides robust results based on a prospectively randomized trial cohort, a potential limitation of the study is the absence of information regarding additional exposures to carcinogens (e.g., occupational or environmental) leading to BC. Additionally, the PLCO trial cohort may be subject to volunteer and differential dropout biases [60]. Nevertheless, the current study reveals the potential of AI in exploring latent patterns from histology images associated with smoker status, thereby helping to examine the hypothesis, as in our case for bladder cancers. According to the evaluation of the MIL approach, we did not find performance benefits compared to the non-MIL approach for our research question, indicating the need to customize the model design according to the study question and the absence of a universal solution. Furthermore, the prediction of the MIL approach for smoke status is random based on the 95% Confidence Interval for AUROC, which was between 0.437 and 0.685. In the in-training validation set, the resection sample from this patient exhibited sparse epithelial components, measuring approximately 0.5 cm in diameter. However, we prioritized the selection of a model that could provide more accurate predictions for patches derived from cases of individuals who had never smoked. It's important to note that the in-training validation set represents a quasi-out-of-distribution compared to the training set. Our primary objective with this validation set is to emulate a one-class detection approach to identify a model that learned histopathologic features associated with individuals who have never smoked. Essentially, we are building a quasi-anomaly detection system assuming that the underlying tumor biology is the primary factor influencing histological appearance. This approach allows us to evaluate the model's ability to identify histological changes linked to smoking as an outlier. Including a class for former smokers as an indeterminate or heterogeneous class in a machine-learning model is a recognized practice. This approach introduces an indeterminate category that can aid the model's regularization. In machine learning, regularization is a technique used to prevent overfitting, where the model performs well on the training data but poorly on new, unseen data. The rationale is based on our assumption that their indeterminate histopathologic characteristics or heterogeneous patterns might facilitate our models to learn the most distinguishable patterns for active and never-smokers. This differentiation can provide more nuanced insights and improve the model's generalization ability for never-smokers and active smokers.

## Conclusion

Histopathologic findings reflected in latent features and exposure scores are associated with the smoking exposure status in bladder cancers. The results of the current study are useful in advancing our understanding of the impact of smoking on bladder cancer pathology.

## Supporting information

**S1 Fig. The flowchart with the case inclusion or exclusion for the current study cohort.** (PNG)

**S2 Fig.** (A) a whole slide image (WSI) with bladder tissue samples; (1) the figure shows the distinctive coloration of hematoxylin-stained nuclei and contrasting appearance of urothelium from less cellular stromal tissue. (2) we applied a color mask to the WSI thumbnails, whose pixels have hue, saturation, and value (HSV) colors located in the color range between (H:140, S:20, V:50) and (H:150, S:255, V:255); (3) Background noise was then filtered from the WSI using the erosion function (kernel size: 2 x 2). Masked areas were expanded, and missing portions were filled with the dilation function (kernel size: 5 x 5); (4) masked areas were split into 10% overlapping squares and subsequently rescaled and remapped to the original WSI size. The masked areas were tiled on WSI at 10x magnification into small patches (512x512 pixels, one pixel = 1 μm) for processing due to memory constraints. (B) illustrates the processing steps on another whole slide image.
(JPEG)

**S3 Fig. PlexusNET Architecture configuration is determined by depth, width, junctions, and blocks; (B) Blocks are composed of various neural network layers. MN: min-max normalization [–1,1], LN: layer normalization. BN: batch normalization. NN: neural networks.**
(PNG)

**S1 Video. This video provides a 3D navigation through the feature space labeled by the tobacco exposure status (yellow: Active smoker, blue: Non-smoker).**
(MOV)

**S1 Table. Lists the overall impression and observation made by two senior uro-pathologists for all subspaces.**
(DOCX)

**S1 File. Provides a list of online access to the entire image patches representing the feature subspaces.**
(DOCX)

## Author Contributions

**Conceptualization:** Okyaz Eminaga, Mahmoud Abbas.

**Data curation:** Okyaz Eminaga.

**Investigation:** Okyaz Eminaga, Hubert Lau.

**Methodology:** Okyaz Eminaga.

**Software:** Okyaz Eminaga.

**Supervision:** Mahmoud Abbas.

**Validation:** Okyaz Eminaga, Hubert Lau, Eugene Shkolyar, Mahmoud Abbas.

**Writing – original draft:** Okyaz Eminaga, Mahmoud Abbas.

**Writing – review & editing:** Hubert Lau, Eugene Shkolyar, Eva Wardelmann, Mahmoud Abbas.

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
