## [Decision Letter · Decision Letter 0]

5 May 2023

PONE-D-23-03953Deep Learning Identifies Histopathologic Changes in Bladder Cancers associated with Smoke Exposure StatusPLOS ONE

Dear Dr. Abbas,

Thank you for submitting your manuscript to PLOS ONE. After careful consideration, we feel that it has merit but does not fully meet PLOS ONE’s publication criteria as it currently stands. Therefore, we invite you to submit a revised version of the manuscript that addresses the points raised during the review process.

We look forward to receiving your revised manuscript.

Kind regards,

Yuchen Qiu, Ph.D.

Academic Editor

PLOS ONE

Journal Requirements:

Reviewers' comments:

Reviewer's Responses to Questions

**Comments to the Author**

1. Is the manuscript technically sound, and do the data support the conclusions?

Reviewer #1: Yes

Reviewer #2: Yes

2. Has the statistical analysis been performed appropriately and rigorously? 

Reviewer #1: Yes

Reviewer #2: Yes

3. Have the authors made all data underlying the findings in their manuscript fully available?

Reviewer #1: No

Reviewer #2: No

4. Is the manuscript presented in an intelligible fashion and written in standard English?

Reviewer #1: Yes

Reviewer #2: Yes

5. Review Comments to the Author

Reviewer #1: Summary

This paper proposed to predict smoke exposure status with histologic changes of BC. A total of

483 whole slide histology images (WSI) of 285 unique cases of BC collected from multiple centers

for BC diagnosis were used in the study. A deep learning model was developed to predict the

smoke exposure status and externally validated on BC cases. The model achieved an AUC of 0.67

in the validation set.

Weakness

1. The AUC(0.64 with continuous parameter and 0.67 with categorized parameter) is rather low

in the validation set. Better than random guess is not sufficient enough to draw the

conclusion that histopathologic features are predictive for smoke exposure status.

2. The authors need a much larger dataset for this study. A total of 367 WSI from 214 cases is

sufficient for developing models for cancer subtypes classification or survival analysis.

However, much more evidence and more convincing results are needed to reveal the

correlation between histologic changes and smoke exposure. Hence a much larger dataset is

necessary.

3. The model is developed under single magnification (10x magnification with a patch size of

512x512 pixels). Why is the model able to capture histologic changes under 10x

magnification? What's the performance under different magnifications(5x, 20x)? I'd like to

see a detailed discussion about the impact of magnification selection on smoke exposure

status prediction.

4. The authors used a CNN architecture for smoke exposure prediction. However, multiple

instance learning (MIL) is more common for WSI analysis[1,2,3], because CNNs tend to

capture local features of WSIs while MILs can represent the global features of WSI. Additional

experiments of MIL and a discussion about the impact of local features and global features

on smoke exposure prediction are recommended for the study.

[1] Campanella, Gabriele, Matthew G. Hanna, Luke Geneslaw, Allen Miraflor, Vitor Werneck Krauss

Silva, Klaus J. Busam, Edi Brogi, Victor E. Reuter, David S. Klimstra, and Thomas J. Fuchs. "Clinicalgrade computational pathology using weakly supervised deep learning on whole slide images."

Nature medicine 25, no. 8 (2019): 1301-1309.

[2] Lu, Ming Y., Drew FK Williamson, Tiffany Y. Chen, Richard J. Chen, Matteo Barbieri, and Faisal

Mahmood. "Data-efficient and weakly supervised computational pathology on whole-slide

images." Nature biomedical engineering 5, no. 6 (2021): 555-570.

[3] Yu, Jin-Gang, Zihao Wu, Yu Ming, Shule Deng, Yuanqing Li, Caifeng Ou, Chunjiang He, Baiye

Wang, Pusheng Zhang, and Yu Wang. "Prototypical multiple instance learning for predicting lymph

node metastasis of breast cancer from whole-slide pathological images." Medical Image Analysis

(2023): 102748.

Reviewer #2: This article use the deep learning model of previous published PlexusNet to distinguish never smoker vs active smoker by analysis patches of whole slice histology images on the PLCO cancer screening trial.

You stated about refs 6~11, “these data are overall limited.” Please explain them in details. And state your novelty.

Please add a purpose section in abstract.

Abstract Results section: “non-randomness” ,why you would like to use “non-randomness”? Do you imply your model is better than a random classifier? I suggest you use another model to do comparison instead stating yours are better than randomness.

Can you draw a figure of data flowchart with inclusion and tons exclusion from 154,900 participants to 1430 BC to 285 cases of development, optimization, and external validation.

Please use consistent terms of development/training, optimization/in-training validation through the paper.

Excluding former smokers from external validation doesn’t make sense. You’re trained and validate your model with three categories including former smokers. Please write the details about how did you transfer three category outputs to two category outputs.

Please draw your deep learning network architecture.

Why did you just use grid search instead of Bayesian Optimization, Hyerband, or random search.

There is an extra space between “age” and “at diagnosis” in the equation of the mixed effect regression.

Please explain why the mixed effect regression equation use exposure-score * cancer-grade instead of exposure-score + cancer-grade.

“All analyses were performed using 2,000-times bootstrap resampling“ I guess you meant you did analysis on the results of external validation set, right?

Table 1, please add the criteria of malignancy grade for readership.

Please use a consistent term of malignancy grade/cancer grade/tumor grade through the paper.

Please give an explanations of both subspace and space. The boundary definition of each subspace is also not clear to me. The t-SNE in Figure 3 didn’t show any clear subspace.

Number of patches of active smoker on in-training validation set was 3(0.09%). Why only 3 patches for this patient?

A typo in discussion. Auerbach(6) is in 1989.

Supplementary Figure 1 need a figure caption.

6. PLOS authors have the option to publish the peer review history of their article (what does this mean?). If published, this will include your full peer review and any attached files.

Reviewer #1: No

Reviewer #2: **Yes: **JINGCHEN MA

---

## [Author Response · Author response to Decision Letter 0]

1 Aug 2023

We would like to express our gratitude to the reviewer for their valuable comments. Based on their feedback, we have made the following edits to the text:

Reviewer #1: 

Summary

This paper proposed to predict smoke exposure status with histologic changes of BC. A total of 483 whole slide histology images (WSI) of 285 unique cases of BC collected from multiple centers for BC diagnosis were used in the study. A deep learning model was developed to predict the smoke exposure status and externally validated on BC cases. The model achieved an AUC of 0.67in the validation set.

Weakness

1. The AUC(0.64 with continuous parameter and 0.67 with categorized parameter) is rather low in the validation set. Better than random guess is not sufficient enough to draw the conclusion that histopathologic features are predictive for smoke exposure status.

Author’s reply:

We appreciate the reviewer for raising this point. According to established statistical conventions [1-5], our analyses possess a robust statistical power of 82.2% at the case level, surpassing the widely accepted threshold of 80% at an alpha level of 0.05. This indicates that the likelihood of our hypothesis being a false positive is only 5%. Therefore, the robustness of our study allows us to derive the conclusion that histopathologic features are likely associated with smoke exposure status. Surely, the accuracy is subject to improvement in the future work. In multivariate mixed effect analyses, we found that the exposure score is the only independent predictor for the active exposure in contrast to BC grade.

We added the following sentence in Statistical analysis section, page 11:

“The statistical robustness of AUC was estimated using the statistical Power. We set the decision threshold for a robust statistical Power to 80% at an alpha level 0.05 (27-31).”

We added the following sentence in the result section, page 13:

“The statistical power of the study was determined to be 82.2%, surpassing the commonly accepted threshold of 80% at an alpha level of 0.05.”

2. The authors need a much larger dataset for this study. A total of 367 WSI from 214 cases is sufficient for developing models for cancer subtypes classification or survival analysis. However, much more evidence and more convincing results are needed to reveal the correlation between histologic changes and smoke exposure. Hence a much larger dataset is necessary.

Author’s reply:

We thank the reviewer once again for bringing up this point. According to our power analyses, we do not see any evidence to support the need for a much larger dataset. In our case, the inclusion of larger datasets may unnecessarily overpower the study and result in highlighting minor differences as significant, according to the common statistical conventions [1-5].

[1] Bacchetti P. Current sample size conventions: flaws, harms, and alternatives. BMC medicine. 2010;8:1-7.

[2] di Stephano J. How much power is enough? Against the development of an arbitrary convention for statistical power calculations. Functional Ecology. 2003;17:707-9.

[3] Greenland S, Senn SJ, Rothman KJ, Carlin JB, Poole C, Goodman SN, et al. Statistical tests, P values, confidence intervals, and power: a guide to misinterpretations. European journal of epidemiology. 2016;31:337-50.

[4] Lipsey MW. Design sensitivity: Statistical power for experimental research: sage; 1990.

[5] Myors B, Murphy KR, Wolach A. Statistical power analysis: A simple and general model for traditional and modern hypothesis tests: Routledge; 2014.

3. The model is developed under single magnification (10x magnification with a patch size of 512x512 pixels). Why is the model able to capture histologic changes under 10x

magnification? What's the performance under different magnifications(5x, 20x)? I'd like to

see a detailed discussion about the impact of magnification selection on smoke exposure

status prediction.

Author’s reply:

We trained a PlexusNET model on 20x magnification. However, we observed performance degradation on the test set (AUROC: 0.528; 95% CI: 0.432 – 0.624 according to DeLong), indicating that magnification does impact performance. Importantly, the association between histology images and smoke exposure status at 20x magnification is random, whereas the association at 10x magnification is significant and not random. This finding indicates that histologic patterns (i.e., tissue architecture) at 10x magnification are predictive of smoke exposure, supporting the key message of the current work regarding the potential association between active smoke exposure and histological images.

We added the following paragraph in the method section, page 8:

“To investigate the effect of the magnification, we repeated the previous steps (except the NAS step; here, we used the resulting model architecture from the first evaluation) to develop and validate a prediction model on images at 20x magnification. After fixing the optimal magnification level, we also examined the attention-based multi-instance learning defined according to Ilse et al. [6] as it has shown performance benefits for certain tasks in digital pathology [7, 8]. The base model was the PlexusNet model determined by NAS that will be described in the following section.”

We added the following paragraph in the result section, page 15:

“When comparing models trained on histology images at 10x magnification versus those trained on images at 20x magnification, we observed performance degradation on the external validation set (AUROC: 0.528; 95% CI: 0.433 – 0.624), indicating that magnification does impact performance.”

We added the following paragraph in the discussion section, page 21:

“The association between histology images and smoke exposure status at 20x magnification is random, whereas the association at 10x magnification is significant and not random. This finding indicates that histologic patterns (i.e., tissue architecture) at 10x magnification are predictive of smoke exposure, supporting the key message of the current work regarding the potential association between active smoke exposure and histological images.“

4. The authors used a CNN architecture for smoke exposure prediction. However, multiple instance learning (MIL) is more common for WSI analysis[1,2,3], because CNNs tend to capture local features of WSIs while MILs can represent the global features of WSI. Additional experiments of MIL and a discussion about the impact of local features and global features on smoke exposure prediction are recommended for the study.

Many papers show the sufficiency of MIL in solving prediction problems.

[1] Campanella, Gabriele, Matthew G. Hanna, Luke Geneslaw, Allen Miraflor, Vitor Werneck Krauss

Silva, Klaus J. Busam, Edi Brogi, Victor E. Reuter, David S. Klimstra, and Thomas J. Fuchs. "Clinicalgrade computational pathology using weakly supervised deep learning on whole slide images."

Nature medicine 25, no. 8 (2019): 1301-1309.

[2] Lu, Ming Y., Drew FK Williamson, Tiffany Y. Chen, Richard J. Chen, Matteo Barbieri, and Faisal

Mahmood. "Data-efficient and weakly supervised computational pathology on whole-slide

images." Nature biomedical engineering 5, no. 6 (2021): 555-570.

[3] Yu, Jin-Gang, Zihao Wu, Yu Ming, Shule Deng, Yuanqing Li, Caifeng Ou, Chunjiang He, Baiye

Wang, Pusheng Zhang, and Yu Wang. "Prototypical multiple instance learning for predicting lymph

node metastasis of breast cancer from whole-slide pathological images." Medical Image Analysis

(2023): 102748.

Author’s reply:

As requested, we considered an Attention-based Multiple Instance Learning (MIL) approach for the model at 10x magnification. However, we did not find performance benefits compared to the non-MIL approach for our research question, indicating the need to customize the model design according to the study question and the absence of a universal solution. Furthermore, the prediction of the MIL approach for smoke status is likely random based on the 95% Confidence Interval for AUROC, which was between 0.437 and 0.685.

In the current study, we further investigated the local features using 3D t-SNE and visualized the 3D feature spaces. We identified global feature subspaces that revealed a high concentration for patches dominated by either active smokers or non-smokers. These findings further indicate reorganization of histologic features associated with these subspaces. Our pathologists investigated and confirmed distinguishable histologic patterns between the subspaces we identified using 3D t-SNE visualization. We emphasize that the 3D visualization of feature spaces provides significantly more information than the 2D visualization of the feature spaces. 

We added the following paragraph in the method section, pages 8-9 :

“After fixing the optimal magnification level, we also examined the attention-based multi-instance learning defined according to Ilse et al. [6] as it has shown performance benefits for certain tasks in digital pathology [7, 8]. The backbone model was the PlexusNet model we defined earlier.”

We added the following paragraph in the result section, page 15:

“In addition to that, MIL failed to provide a non-random prediction for the smoke exposure status (AUROC: 0.561; 95% CI: 0.437 – 0.685).”

We added the following paragraph in the discussion section, pages 21 and 22:

“In the current study, we further investigated the local features using 3D t-SNE and visualized the 3D feature spaces. We identified global feature subspaces that revealed a high concentration for patches dominated by either active smokers or non-smokers. These findings further indicate reorganization of histologic features associated with these subspaces. Our pathologists investigated and confirmed distinguishable histologic patterns between the subspaces we identified using 3D t-SNE visualization. We emphasize that the 3D visualization of feature spaces provides significantly more information than the 2D visualization of the feature spaces.”

“According to the evaluation of the MIL approach, we did not find performance benefits compared to the non-MIL approach for our research question, indicating the need to customize the model design according to the study question and the absence of a universal solution. Furthermore, the prediction of the MIL approach for smoke status is random based on the 95% Confidence Interval for AUROC, which was between 0.437 and 0.685.”

Reviewer #2: 

This article use the deep learning model of previous published PlexusNet to distinguish never smoker vs active smoker by analysis patches of whole slice histology images on the PLCO cancer screening trial.

1. You stated about refs 6~11, “these data are overall limited.” Please explain them in detail. And state your novelty.

Author’s reply:

We do not know whether histologic appearance in histology images is associated with smoke exposure. The current work is the first study to investigate this association.

We modified the following section in the introduction, page 3:

“However, the existing data from previous studies are characterized by overall inconsistencies and a focus on the controversial association between BC grade and smoke exposure. As a result, it remains unclear whether there is indeed an association between specific histologic features in BC and smoke exposure. Given this knowledge gap, the current study investigated the potential association between histology images and smoke exposure status.”

2. Please add a purpose section in abstract.

Author’s reply:

Added accordingly in the abstract.

“Given this knowledge gap, the current study investigated the potential association between histology images and smoke exposure status.”

3. Abstract Results section: “non-randomness” ,why you would like to use “non-randomness”? Do you imply your model is better than a random classifier? I suggest you use another model to do comparison instead stating yours are better than randomness.

Author’s reply:

In statistical terms, "non-randomness" refers to the presence of significant non-random associations between histology and active smoke exposure status. The main aim of this work is to reveal the association between histology images and smoke exposure using deep learning. We also used BC grade as a reference variable, which was not associated with smoke exposure, in contrast to our smoke score.

4. Can you draw a figure of data flowchart with inclusion and tons exclusion from 154,900 participants to 1430 BC to 285 cases of development, optimization, and external validation.

Authors’ reply:

We appreciate the suggestion to provide a flowchart illustrating the data flow. We have included a supplementary figure in the paper to illustrate the process, showing the inclusion and exclusion of participants from 154,900 to 1,430 breast cancer cases, and further narrowing down to 285 cases for development, optimization, and external validation.

5. Please use consistent terms of development/training, optimization/in-training validation through the paper.

Authors’ reply:

We have revised the text to ensure consistent use of terms such as training and optimization/in-training validation throughout the paper. We used the term development set to covers both the training set and the in-training validation/optimization set.

6, Excluding former smokers from external validation doesn’t make sense. You’re trained and validate your model with three categories including former smokers. Please write the details about how did you transfer three category outputs to two category outputs.

Authors’ reply:

We disregarded the classification prediction for former smokers and only considered the prediction score for active smokers for the association analyses. The primary reason for including the class for former smokers during model development is to regularize the model prediction. However, considering the former smoker status for external validation is not appropriate given the difficulty in identifying the regeneration level of the tissues resulting from stopping smoke exposure. In contrast, we can assume that the baseline environmental exposure (including passive smoking) affects both smokers and non-smokers equally, while smokers have direct exposure to tobacco. 

Accordingly, we added the previous paragraph in the discussion section, pages 20 and 21.

7. Please draw your deep learning network architecture.

Authors’ reply:

We appreciate the reviewer's request to provide the model architecture. However, instead of drawing the network architecture, we utilized a recently published model architecture concept that emphasizes the complexity of visualization. We believe this approach will be more beneficial for readers to understand the model's structure and functionality.

8. Why did you just use grid search instead of Bayesian Optimization, Hyerband, or random search.

Authors’ reply:

The search space for architecture hyperparameters is finite, limited, and searchable using the conventional grid search. The computation cost for grid search is also reasonable. 

Accordingly, we added the following sentences in the Method section, page 7:

“The grid search was preferred because the search space for architecture hyperparameters was finite, limited, and searchable using the conventional grid search. The computation cost for grid search was also reasonable.”

9. There is an extra space between “age” and “at diagnosis” in the equation of the mixed effect regression.

Authors’ reply:

Corrected accordingly.

10. Please explain why the mixed effect regression equation use exposure-score * cancer-grade instead of exposure-score + cancer-grade.

Authors’ reply:

Both parameters are derived from histology appearance, and we anticipated the existence of interaction between them. From statistical perspective, we applied this to achieve a non-singular fit in contrast to exposure-score + cancer-grade that resulted in a singular fit when the random effect including gender and age at diagnosis is considered.

We added the following sentences in M&M section, page 9:

“The random effect incorporated gender and age. To avoid a singular model fit resulting from the limited random effect, we defined the interaction between exposure

---

## [Decision Letter · Decision Letter 1]

9 Jan 2024

PONE-D-23-03953R1Deep Learning Identifies Histopathologic Changes in Bladder Cancers associated with Smoke Exposure StatusPLOS ONE

Dear Dr. Abbas,

Thank you for submitting your manuscript to PLOS ONE. After careful consideration, we feel that it has merit but does not fully meet PLOS ONE’s publication criteria as it currently stands. Therefore, we invite you to submit a revised version of the manuscript that addresses the points raised during the review process.

We look forward to receiving your revised manuscript.

Kind regards,

Yuchen Qiu, Ph.D.

Academic Editor

PLOS ONE

Reviewers' comments:

Reviewer's Responses to Questions

**Comments to the Author**

1. If the authors have adequately addressed your comments raised in a previous round of review and you feel that this manuscript is now acceptable for publication, you may indicate that here to bypass the “Comments to the Author” section, enter your conflict of interest statement in the “Confidential to Editor” section, and submit your "Accept" recommendation.

Reviewer #1: (No Response)

Reviewer #3: (No Response)

2. Is the manuscript technically sound, and do the data support the conclusions?

Reviewer #1: Partly

Reviewer #3: Partly

3. Has the statistical analysis been performed appropriately and rigorously? 

Reviewer #1: Yes

Reviewer #3: No

4. Have the authors made all data underlying the findings in their manuscript fully available?

Reviewer #1: Yes

Reviewer #3: Yes

5. Is the manuscript presented in an intelligible fashion and written in standard English?

Reviewer #1: Yes

Reviewer #3: Yes

6. Review Comments to the Author

Reviewer #1: In this study, The authors developed a deep learning model to assess the association between cigarette smoking and histopathologic changes in bladder cancer by analyzing morphological features in pathology slides. The authors found that the deep learning model can extract smoking-related histologic features from slides and generated exposure score to predict smoking status using mixed-effect model. The model exhibited moderate ability to distinguish smoking status, suggesting smoking may induce specific pathologic changes in bladder cancer.

Overall, the finding of manuscript is clear and easy to read. However, I am concerned about the model development process. Here are some questions.

1. The authors excluded former smokers from external validation cohort to eliminate the potential effect of smoking cessation on histopathological appearance. For this reason, why were former smokers not also excluded from the training cohort?

2. The authors utilized neural architecture search to construct the network. However, as a computer vision task in digital pathology, why not use well-established pretraining model such as ViT and ResNet pretrained on natural images or CTransPath pretrained on WSI so that the model could have better image representations.

3. The cohort 'In-training validation set' only contains 1 activate smoker. I doubt whether the model performance could be evaluated properly during model development with such an imbalanced validation set.

4. "Discriminatory accuracy was determined by classification accuracy, specificity, sensitivity, f1-score, recall, precision, negative and positive predictive values."

Sensitivity and recall are identical; precision and positive predictive value are identical. The expression should be clarified in the manuscript. And in the Results section, "sensitivity of 82% (95% CI: 71 – 93)" and "recall rate of 0.82 (95% CI: 0.69 – 0.91)", these values ought to be the same. Why is there a difference.

5. "The base model considered gender and age at diagnosis as random effects, and malignancy grade and time to diagnosis as fixed effects."

Why gender and age at diagnosis were considered as random effects while malignancy grade and time to diagnosis were fixed effects? Should be clarified.

6. The variable "time to diagnosis" should be further clarified. What is the start time of the variable? Why is it correlated with smoke status?

7. How is "the general equation of the mixed effect regression model" designed? What are definitions of the "•", "||" and "()" in the equation?

Reviewer #3: The author has addressed several concerns from the last round of the review. However, some of the questions have not been fully addressed. The remaining questions are listed below:

(1) The classification performance (0.67 AUC) is still not strong enough to support the reliability of the latent space features provided by the model. The author only reports the performance of one MIL method, which still lacks sufficient comparison with baseline models. More baseline methods with grid search are needed to demonstrate the capability of the backbone selected in this paper.

(2) The author claimed that "The primary reason for including the class for former smokers during model development is to regularize the model prediction." Please provide an ablation study with the model trained (1) with former smokers and (2) without former smokers to demonstrate the benefit of including former smokers during the training process.

(3) The testing is not consistent with the training. Even the external validation only has never/active smokers. A multi-class AUC is required to report what happens if the model predicts one of the external validation cases as former smokers based on the exposure scores, neither never nor active. Why is the threshold set at 65?

(4) Since the author didn't perform NAS for different magnifications (5X, 10X, 20X), the selection of the optimal magnification is not reliable. More comprehensive investigations with different backbones are required to draw a conclusion.

(5) If the data only has patient-level labels, how do you separate the latent-space features at the patch level in the 3D t-SNE figures? How do you locate those three subspaces? How can you ensure that those three subspaces can represent the main features of the whole dataset? Are there any other subspaces that need to be examined?

(6) Since the size of the dataset is limited, K-fold cross-validation with a larger value of K is required. The current setting in the paper with K=2 might not be sufficient.

7. PLOS authors have the option to publish the peer review history of their article (what does this mean?). If published, this will include your full peer review and any attached files.

Reviewer #1: **Yes: **Yan, Ziye

Reviewer #3: No

---

## [Author Response · Author response to Decision Letter 1]

17 Jan 2024

Response to Reviewer #1:

1. Inclusion of Former Smokers in the Training Cohort: We recognize the reviewer's concerns about including former smokers in our training cohort. Adopting a category for former smokers as an indeterminate class is a recognized approach in machine learning. This practice introduces an indeterminate category which contributes to model regularization, a strategy aimed at preventing overfitting. Overfitting happens when a model is highly accurate on training data but performs poorly on new, unseen data. Our rationale for incorporating former smokers into the training dataset is based on the premise that their indeterminate histopathologic features and heterogenous patterns could enhance the model's ability to accurately distinguish between active smokers and never-smokers. This approach hopefully mitigates the model from learning indeterminate features found in the active smokers or never-smokers. 

For testing, we opted to exclude former smokers due to the complexities involved in assessing tissue regeneration after smoking cessation. The inclusion of former smokers in our test analysis could potentially skew the results, considering their intermediate status blurs the lines between active and never smokers.

2. Choice of Neural Architecture Search over Established Models: Our research primarily focused on examining the relationship between smoking and histopathologic changes, rather than enhancing machine learning models. We employed NAS + PlexusNET, a well-examined neural network framework on across five medical image modalities by a previous study, comparing it to the Multiple Instance Learning (MIL) approach that reflects the global context learning. We also evaluated the models at 10x and 20x magnifications to understand the impact of magnification on histology-tobacco correlation.

While integrating models like ViT, ResNet, or CTransPath could have been beneficial, it was beyond our study's immediate scope. Previous research demonstrated that PlexusNET offers computational efficiency and competitive results in various medical imaging modalities. This indicates that our NAS-driven approach balances accuracy and efficiency better than traditional “established” models.

The Vision Transformer (ViT) model requires more computing power compared to PlexusNET and many typical CNN architectures. On the other hand, CTransPath uses CNN to detect local granular structures. Our PlexusNET, primarily a CNN model, is designed to identify fine granular structure associate to tobacco exposure. Additionally, PlexusNET can adopt a hybrid approach, combining CNN with Transformer, as optimized by Neural Architecture Search (NAS), for improved performance in capturing detailed patterns. Overall, in-depth discussion would be beyond the scope of the current study.

3. Imbalanced In-Training Validation Set: Our in-training validation set, which originated from a single center independent of both the training and test sets, exhibited a limited representation of active smokers but included a sufficient representation of former smokers. Our primary objective with this validation set is to emulate a one-class detection approach, aiming to identify a model that learned histopathologic features associated with individuals who have never smoked. Essentially, we are building a quasi-anomaly detection system, under the assumption that the underlying tumor biology is the primary factor influencing histological appearance. This approach allows us to evaluate the model's capability to identify histological changes linked to smoking as outlier.

4. Clarification of Statistical Terms: The variations in the confidence intervals for sensitivity and recall rates arise due to the application of bootstrapping methods with distinct seeds. This can lead to minor differences in the confidence intervals, even though the underlying measures remain the same. To avoid potential confusion for readers, we removed recall from the main text.

5. Random vs. Fixed Effects in the Model: In our analysis, we treated gender and age as random effects, recognizing them as demographic variables that were distributed randomly within the Prostate, Lung, Colorectal, and Ovarian (PLCO) Cancer Screening Trial cohort. This approach acknowledges the inherent variability in these factors across the population. Conversely, we designated time to diagnosis and malignancy grade as fixed effects. This decision aligns with the prospective nature of the PLCO study, where the time to diagnosis was meticulously recorded. We postulate that the time to diagnosis encompasses the latent period leading up to the clinical manifestation of bladder cancer. Furthermore, we suggest that the malignancy grade correlates with the smoking status of the individuals in the study. This correlation potentially extends to an indirect influence by the latent period and the duration of tobacco exposure within the time-to-diagnosis frame.

6. Clarification on “Time to Diagnosis”: The variable 'time to diagnosis' was derived from the PLCO study and represents the duration from the participants' inclusion in the screening study. We examined its correlation with smoking status, considering the length of exposure. Additionally, we assumed that 'time to diagnosis' might also be related to any latent period in the development of clinically visible bladder cancer, which could, in turn, influence the malignancy grade. This assumption is based on findings in the literature that suggest a connection between malignancy grade and smoking status.

7. Explanation of Mixed Effect Regression Model Symbols: We have added clarification in our manuscript regarding the symbols used in the mixed-effect regression model. The symbols represent interaction (.), conditional relations (||), and grouping (()). These are standard notations in statistical modeling.

Response to Reviewer #3:

1. Classification Performance and Reliability: We appreciate the reviewer's concerns regarding the AUC of 0.67. However, it's essential to provide clarity on the primary objective of our study, which was to explore the association between histology and smoking status, rather than solely aiming for the highest classification performance.

We would like to reiterate, as mentioned in our initial revision letter, that our approach is in alignment with standard statistical practices. It's important to understand that modeling can serve two primary purposes: investigating associations or making predictions (e.g. classification). In our study, we focused on examining associations rather than predictions, and the study design was accordingly tailored to this objective.

We would like to highlight the robust statistical power of our study, with a confidence level of 82.2% at a 5% alpha level. This statistical strength reinforces the credibility and reliability of our findings, especially within the specific scope of our research objectives.

Moreover, we firmly believe that our assessment is both objective and well-founded. It aligns with a prior study (cited in the main text) that also hinted at an association between tobacco exposure and histopathological changes that were not considered for malignancy grading. Additionally, multiple other studies have indicated the existence of a potential relationship between malignancy grades and tobacco exposure in bladder cancer (already referenced in the main text).

In consideration of these compelling factors, we firmly believe that our study contributes valuable insights to the field. It sheds light on the intriguing connection between histopathology and tobacco exposure, which we anticipate will ignite further evaluation of the impact of environmental or human factors on cancer histology, opening new avenues for research and understanding in this domain.

2. Rationale for Including Former Smokers: Including a class for former smokers as indeterminate or heterogenous class in a machine learning model is a recognized practice. This approach introduces an indeterminate category that can aid in the regularization of the model. Regularization, in machine learning, is a technique used to prevent overfitting, where the model performs well on the training data but poorly on new, unseen data. The rationale is based on our assumption that their indeterminate histopathologic characteristics or heterogenous patterns might facilitate our models to learn most distinguishable patterns for the active smokers and never-smokers. This differentiation can provide more nuanced insights and improve the model's ability to generalize for never-smoker and active smokers.

Ablation studies are useful for understanding the impact of certain features or categories on the model's performance, but they can also be resource-intensive and does not align with the specific goal of our study which is to investigate the association between histology images and smoke exposure.

Our approach, focusing on a training set that includes a broad spectrum of cases (including former smokers) and a highly curated test set, is aimed at ensuring that the model is capable to detect association between tobacco exposure (active vs never) and histology. This methodology is consistent with best practices in machine learning, where diverse training data helps in building robust models, and a well-curated test set is crucial for accurately assessment.

3. Consistency in Testing and Training: The use of a two-class system in our test set (active vs. never smokers) was intentional, aligning with our research objectives and already justified in the previous points. The threshold of 65 was determined using the median value on the development set and fixed for the test set, which is a standard approach in such analyses. We highlighted the following sentence related to thresholding (highlighted in green in the main text, page 8).

“Afterward, the median exposure score was estimated from the development data set and used as the threshold for conversion of the exposure score from a continuous to binarily categorized parameter, and this was locked as the threshold for the external validation set.”

4. Optimal Magnification Selection: While comprehensive investigations across different magnifications might be informative, our study did include the investigation on 10x and 20x objective magnification. Model resulting from NAS was identified on development set with 32x32 pixels downsized images following the instruction published earlier, which we think to be reliable for our study’s purpose, minimizes the impact of magnification on the model selection. We refer the reviewer to our previous work that thoroughly investigated our NAS strategy on 5 different medical imaging modalities including pathology images.

In summary, NAS was not utilized for either 10x or 20x magnification, rendering the concerns unfounded.

5. 3D t-SNE Visualization and Feature Representation: The 3D t-SNE visualization serves as a tool for data exploration and facilitates the understanding of feature representation by visualizing them in a manner that is browsable and readable for humans to identify subspaces relevant for the study question. We labelled the data points according to the smoke status and exposure score to visually identify the subspaces we described in the main text as well while ensuring a sophisticated case coverage in these subspaces. Our pathologists have confirmed the distinct histologic patterns observed in the identified feature subspaces we described in the main text, and we provided multiple images originated from these subspaces to validate our perspective in the supplementary material section. We refer the reviewer for further information regarding the t-SNE visualization as it is beyond the scope of the current study to investigate the limitations of t-SNE or to provide alternative solutions.

6. Justification of K-fold Cross-Validation Setting: Our decision to use K=2 in cross-validation during NAS was based on previously established rationales in our earlier work which the computational efficacy and the completion of NAS within 24 hours using a single GPU. We showed that our NAS approach using 2-fold CV deliver models that are on par and superior to existing SOTA models. Moreover, expanding on this is beyond the current study’s scope. Accordingly, we refer the reviewer to our earlier work (https://pubmed.ncbi.nlm.nih.gov/36753979/) that justified the use of the 2-fold cross validation.

---

## [Decision Letter · Decision Letter 2]

11 Apr 2024

PONE-D-23-03953R2Deep Learning Identifies Histopathologic Changes in Bladder Cancers associated with Smoke Exposure StatusPLOS ONE

Dear Dr. Abbas,

Thank you for submitting your manuscript to PLOS ONE. After careful consideration, we feel that it has merit but does not fully meet PLOS ONE’s publication criteria as it currently stands. Therefore, we invite you to submit a revised version of the manuscript that addresses the points raised during the review process.

We look forward to receiving your revised manuscript.

Kind regards,

Yuchen Qiu, Ph.D.

Academic Editor

PLOS ONE

Journal Requirements:

Reviewers' comments:

Reviewer's Responses to Questions

**Comments to the Author**

1. If the authors have adequately addressed your comments raised in a previous round of review and you feel that this manuscript is now acceptable for publication, you may indicate that here to bypass the “Comments to the Author” section, enter your conflict of interest statement in the “Confidential to Editor” section, and submit your "Accept" recommendation.

Reviewer #1: All comments have been addressed

2. Is the manuscript technically sound, and do the data support the conclusions?

Reviewer #1: Yes

3. Has the statistical analysis been performed appropriately and rigorously? 

Reviewer #1: Yes

4. Have the authors made all data underlying the findings in their manuscript fully available?

Reviewer #1: Yes

5. Is the manuscript presented in an intelligible fashion and written in standard English?

Reviewer #1: Yes

6. Review Comments to the Author

Reviewer #1: The reserch is intresting and it is maybe decover the relationship between the smoking and the histopathologic changes.

Only the result of deep learning model is given, the result of baseline method is absent.

The details of the model is necessary, such as a charflow or a network frame diagram.

7. PLOS authors have the option to publish the peer review history of their article (what does this mean?). If published, this will include your full peer review and any attached files.

Reviewer #1: **Yes: **Yan, Ziye

---

## [Author Response · Author response to Decision Letter 2]

26 Apr 2024

We thank the reviewer for the valuable feedback.

Reviewer #1: 

The reserch is intresting and it is maybe decover the relationship between the smoking and the histopathologic changes.

Authors’ reply:

We appreciate the positive feedback and recognition of the relevance of our work.

Only the result of deep learning model is given, the result of baseline method is absent.

Authors’ reply:

Given that Grade is not a significant independent predictor of smoke exposure in the multivariate mixed-effect logistic regression, there is no point in providing the performance in AUC (as a measurement of randomness) or F1 score (as a measurement of accuracy) of the baseline method. We already provided MIL performance as a comparison on page 16, line 4:

“In addition, MIL failed to provide a non-random prediction for smoke exposure status (AUROC: 0.561; 95% CI: 0.437 – 0.685).” Accordingly, given the randomness of prediction for MIL, there was no point in providing an F1 score here.

The details of the model is necessary, such as a charflow or a network frame diagram.

Authors’ reply:

We provided a model architecture configuration following definitions published in the previous paper (https://pubmed.ncbi.nlm.nih.gov/36753979/). Additionally, we included the PlexusNET general network concept as supplementary Figure 3.

---

## [Editor Report · Decision Letter 3]

24 May 2024

Deep Learning Identifies Histopathologic Changes in Bladder Cancers associated with Smoke Exposure Status

PONE-D-23-03953R3

Dear Dr. Abbas,

We’re pleased to inform you that your manuscript has been judged scientifically suitable for publication and will be formally accepted for publication once it meets all outstanding technical requirements.

Kind regards,

Yuchen Qiu, Ph.D.

Academic Editor

PLOS ONE

---

## [Editor Report · Acceptance letter]

24 Jun 2024

PONE-D-23-03953R3 

PLOS ONE

Dear Dr. Abbas, 

I'm pleased to inform you that your manuscript has been deemed suitable for publication in PLOS ONE. Congratulations! Your manuscript is now being handed over to our production team.

Kind regards, 

on behalf of

Dr. Yuchen Qiu 

Academic Editor

PLOS ONE